# Lignin as a High-Value Bioaditive in 3D-DLP Printable Acrylic Resins and Polyaniline Conductive Composite

**DOI:** 10.3390/polym14194164

**Published:** 2022-10-04

**Authors:** Goretti Arias-Ferreiro, Aurora Lasagabáster-Latorre, Ana Ares-Pernas, Pablo Ligero, Sandra María García-Garabal, María Sonia Dopico-García, María-José Abad

**Affiliations:** 1Grupo de Polímeros-CITENI, Campus Industrial de Ferrol, Universidade da Coruña, Campus de Esteiro, 15403 Ferrol, Spain; 2Departamento Química Orgánica I, Facultad de Óptica, Universidad Complutense de Madrid, Arcos de Jalón 118, 28037 Madrid, Spain; 3Enxeñería Química Ambiental Group, Centro de Investigacións Científicas Avanzadas (CICA), Universidade da Coruña, 15071 A Coruña, Spain; 4Grupo Mesturas, Universidade da Coruña, Campus da Zapateira s/n, 15071 A Coruña, Spain

**Keywords:** lignin, DLP, polyaniline, filler dispersibility, additive manufacturing, acrylic resin

## Abstract

With increasing environmental awareness, lignin will play a key role in the transition from the traditional materials industry towards sustainability and Industry 4.0, boosting the development of functional eco-friendly composites for future electronic devices. In this work, a detailed study of the effect of unmodified lignin on 3D printed light-curable acrylic composites was performed up to 4 wt.%. Lignin ratios below 3 wt.% could be easily and reproducibly printed on a digital light processing (DLP) printer, maintaining the flexibility and thermal stability of the pristine resin. These low lignin contents lead to 3D printed composites with smoother surfaces, improved hardness (Shore A increase ~5%), and higher wettability (contact angles decrease ~19.5%). Finally, 1 wt.% lignin was added into 3D printed acrylic resins containing 5 wt.% p-toluensulfonic doped polyaniline (pTSA-PANI). The lignin/pTSA-PANI/acrylic composite showed a clear improvement in the dispersion of the conductive filler, reducing the average surface roughness (R_a_) by 61% and increasing the electrical conductivity by an order of magnitude (up to 10^−6^ S cm^−1^) compared to lignin free PANI composites. Thus, incorporating organosolv lignin from wood industry wastes as raw material into 3D printed photocurable resins represents a simple, low-cost potential application for the design of novel high-valued, bio-based products.

## 1. Introduction

The development of 3D printing technologies requires new materials that are environmentally friendly and able to provide new features, specifically to increase the range of applications of vat polymerization-based techniques [1,2,3]. New 3D printing technologies are becoming important as production models in the frame of Industry 4.0 and the Internet of Things [4,5]. Stereolithography (SLA) and digital light processing (DLP) are vat polymerization printing techniques based on the use of UV-light to solidify a liquid resin layer-by-layer [6]. To achieve successful printing, the photo-rheological and mechanical properties of the multicomponent resins need to be adjusted [7]. These photosensitive resins are complex mixtures of a photoinitiator, monomers, oligomers, and other additives intended to tune their properties [6].

In this context, lignin can play an important role in the development of new resin formulations as the second most abundant natural polymer on the planet, low cost, and renewable [8,9,10]. Most lignin is obtained as a by-product of the wood industry and valorized energetically as a fuel. With increasing environmental awareness and the circular economy, the use of lignin has been reconsidered to look for its application in high-valued bio-based products [11,12,13]. Lignin is a reticular polyphenolic polymer with a three-dimensional structure of high structural complexity, amorphous, and very heterogeneous. Its structure and final properties are directly related to both its natural origin and the treatment followed for its extraction and purification, such as organosolv, soda, or kraft process [10,14]. Within polymers, lignin has a wide range of applications such as additive to enhance their mechanical behavior (hardness, elasticity and tensile strength), provide antioxidant properties, tune the surface wettability, improve biodegradability [14], or as compatibilizer in polymer composites [15].

In the field of electronics, lignin has been widely investigated [11,12,16,17,18,19,20,21,22] for the manufacture of energy storage devices [22], electromagnetic shielding [16,17], organic cathodes [11], as a natural binder or even as a dopant in conductive polymers for the production of electrochemical capacitors or supercapacitors [12,18,19,20,21]. Due to the redox activity of the quinone/hydroquinone moieties, lignin derivatives effectively enhanced the capacitance of intrinsically conductive polymers, including poly(3,4-ethylenedioxythiophene), polypyrrole, and polyaniline (PANI) [23] as well as carbon-based materials such as graphene [19]. Sulfomethylated lignin has been shown to effectively dope PANI and increase the conductivity and dispersibility of lignin/PANI composites in water [24].

In this context, future electronic devices could provide great benefits from the synergistic combination of AM techniques and green chemistry, delivering a wider range of 3D printed polymer composites with improved electrochemical performance and enhanced physicochemical properties. Nowadays, although lignin has been employed in a variety of 3D printing applications, its use in vat polymerization is still scarce [7,25].

In order to add lignin in SLA/DLP resins two strategies have been investigated: directly mixing unmodified lignin and functionalized bulk lignin with the photoreactive monomers formulation. The blending of unmodified kraft lignin with methacrylate resin improved the mechanical properties of the printed samples to a limited extent, as amounts higher than 1% lignin seriously hindered the crosslinking reaction [26]. Further, the ability of small organosolv lignin loadings as a compatibilizer for graphene nanoplatelets (G) has been positively proven in the mechanical properties enhancement of photocurable polyurethane–Lignin/G [27]. On the other hand, Sutton et al. [28] incorporated up to 15 wt.% of organosolv lignin functionalized with methacrylic anhydride in a commercial SLA resin, increasing the ductility of the materials due to the plasticizing effect of the lignin side chains [28].

In a previous work, the authors developed a formulation based on the conductive filler PANI-lignin in a photocurable acrylic matrix, which proved to be valid for the manufacture of flexible wearable electronics and sensors [23]. The present research is focused on the incorporation of unmodified organosolv lignin as an additive to the photocurable resin. The purpose is to deepen the composite behavior and explore other potential applications of lignin in the manufacture of functional materials, such as the dispersant of conductive fillers in composites for electronic devices. We present a study of the structure-property relationship of 3D printed composites based on organosolv lignin isolated from Betula alba bark and a home-made photocurable acrylic matrix. Lignin was used as a non-reactive filler without any further chemical modification.

In a preliminary stage, the resin curing properties were assessed through viscosity measurements, real-time Fourier transform infrared spectroscopy, and the Jacob’s working curve approach. Then, the 3D printed lignin composites were thoroughly characterized by SEM, FTIR, TGA, tensile properties, Shore D hardness, and water contact angle. As way of example, the combination of lignin and para-toluene sulfonic acid doped polyaniline (pTSA-PANI) lead to an improvement of the filler dispersion within the acrylic matrix, resulting in more homogeneous samples with lower surface roughness and enhanced electrical conductivity.

## 2. Experimental Section/Methods

### 2.1. Materials and Samples Preparation

Aniline (ANI, 99.5%) and para-toluene sulfonic acid (pTSA, 98%) were supplied by Sigma-Aldrich (St. Louis, MO, USA) and ammonium persulfate (APS, 99%) was obtained from Acros (Geel, Belgium). Monomer, crosslinker and photoinitiator were ethylene glycol phenyl ether acrylate (EGPEA, molecular weight = 192.21 g/mol), 1,6-hexanediol diacrylate (HDODA, molecular weight = 226.27 g/mol), and diphenyl (2,4,6-trimethylbenzoyl) phosphine oxide (TPO, molecular weight = 348.37 g/mol), respectively; they were obtained from Sigma-Aldrich (St. Louis, MO, USA). Acetone, chloroform, glycerin, methanol and 2-propanol were purchased from Scharlau (Sentmenat, Spain).

***Preparation of Lignin.*** Lignin was extracted from Betula Alba bark by organosolv fractionation using as a solvent acetic acid and hydrochloric acid, as described in a previous reference [23]. The obtained lignin was lyophilized before its use.

The characterization of the molecular weight distribution of lignin was performed by gel permeation chromatography (GPC). A Waters 2695 system (Waters, Mildford, MA, USA) was used, equipped with two linear columns (Styragel 4E and Styragel HR3, 4.6 × 300 mm) and an ultraviolet diode array detector (PDA, model 996 UV); the temperature of the column was 35 °C and the mobile phase tetrahydrofuran (THF) at 0.3 mL min^−1^. A calibration curve obtained with polystyrene standards in the range 580 to 93,800 Da was used (220 nm). Samples of lignin were dissolved in the mobile phase with a concentration of 1000 mg/L by gently stirring the samples up to 24 h. The values obtained were Mw = 2115, Mn = 924 and polydispersity = 2.3 (average of 3 replicates).

***PANI******synthesis.*** PANI was synthesized adapting the emulsion polymerization method described by Dopico et al. for the synthesis of PANI doped with dodecyl benzenesulfonic acid (DBSA-PANI) in a mixture of water and chloroform, changing the dopant to pTSA [29]. Aniline hydrochloride (2 mL, 1.88 g) was dissolved in 200 mL chloroform. The oxidant ammonium peroxydisulfate (APS, 4.5 g) and para-toluene sulfonic acid (pTSA, 11.4 g) were dissolved in 100 mL water. The polymerization reaction was initiated by adding dropwise the aqueous solution to the aniline solution and allowed to proceed for 24 h under mechanic stirring (150 rpm) and low temperature (<6 °C). The resulting product was collected through precipitation with methanol and acetone and filtered under vacuum. pTSA doped polyaniline (pTSA-PANI) was obtained as a very dark green powder after drying during 24 h at 40 °C in a vacuum oven. The yield recovered was 2.39 g. The elemental analysis of pTSA-PANI was performed in duplicate on a ThermoFinnigan Flash EA1112 analyzer. The average result was: 8.34 wt.% N; 51.74% C; 4.74% H; 9.52% S.

***Preparation and printing of the composites.*** Samples were formulated taking as reference the base resin of previous studies [30,31]. Fixed quantities of the HDODA crosslinker (15 wt.%) and the TPO photoinitiator (7 wt.%) were mixed with the EGPEA, as a major monomer. Increasing amounts of unmodified lignin were then added. The formulations of the lignin 3D printing composites analyzed in this investigation are presented in Table 1. In order to evaluate the effect of lignin as a dispersant of conductive charges in functional materials, two additional formulations were prepared using 5 wt.% pTSA-PANI with 1 wt.% lignin (LG1PANI5) and without lignin (LG0PANI5) as a control.

To promote homogeneous dispersion each sample formulation was sonicated during 30 min using a Digital Sonifier at 10% of intensity (Branson 450, Danbury, CT, USA) and further mechanically stirred in a Vortex mixer for 2 min at 1000 rpm (VELP Scientific, Schwabach, Germany) just prior printing. The 3D DLP printer was an Elegoo mars PRO (wavelength 405 nm; 9 mW.cm^−2^); its settings were adjusted considering the printer technical requirements and the kinetics of polymerization of the formulations tested. The exposure time per layer was adjusted for each formulation (Table 1). Bottom exposure corresponds to the exposure time of the first 5 layers, which is always longer to ensure suitable adhesion of the sample to the metal platform. The layer thickness (z) was set at 0.025 mm. The printability worsens from 3% lignin composites which can be explained by their longer curing times and adhesion problems to the printing platform. 

The uncured resin remaining in the printed samples was removed by soaking them in 2-propanol for 10 min. A post-curing process was performed using a lamp (Form Cure, Formlabs) for 5 min at 35 °C. The sample characterization was carried out on flexible dog-bone-shaped specimens according to ISO 527. The preparation of the samples for 3D printing and an example of the printed samples are shown in Figure 1.

### 2.2. Sample Characterization

The viscosity of the acrylic composite formulations was measured using a controlled strain rheometer (ARES, TA Instruments, Newcastle, DE, USA) with parallel-plate geometry (25 mm diameter, 1 mm gap) at room temperature. Steady shear viscosity (η) was measured in the range of shear rates 0.3–100 s^−1^.

The UV-Vis spectra of 4 different suspensions of lignin and pTSA-PANI in glycerin (spanning from 50 to 200 ppm) were recorded on a Jasco V-750 double-beam UV-Vis spectrophotometer (Jasco Analítica S.L., Madrid, Spain) between 330 and 800 nm with a sampling interval of 1 nm and 25 accumulations.

The kinetics of the UV-initiated radical polymerization of the acrylic composites with 0, 1, 2, 3, and 4 wt.% lignin were studied in situ by Real Time FTIR spectroscopy (Jasco 4700 spectrometer, Jasco Analítica S.L., Madrid, Spain) in Attenuated Reflectance Mode (ATR) (MK II Golden Gate^TM^ Diamond 45° ATR). The details of the full procedure are included in a previous reference [32]. The analyses were performed in triplicate with a 4 cm^−1^ resolution between 1800 and 550 cm^−1^ over 5 scans (6 s). A Visicure 405 nm spot lamp connected to an LED Spot-Curing System (BlueWave, Dymax Corp., Torrington, CT, USA), was used to cure the sample from its top side. Bruker OPUS^®^ software version 5.5 (Bruker Española S.A, Madrid, Spain) was employed for the spectra analysis.

The Jacobs working curves of the liquid formulations were calculated from samples cured over the screen of the 3D printer, varying the exposure times; circular films of 18 mm in diameter and 1 mm thickness were printed, followed by a clean-up step with 2-propanol. The thickness of the samples was measured with a thickness DUALSCOPE^®^ MP0R measuring instrument (Fisher) and plotted as a function of Exposure (*E*_*max*_) (mJ.cm^−2^). Each sample was performed in duplicate.

The morphology of lignin and pTSA-PANI was evaluated by Transmission Electron Microscopy (TEM) (JEOL JEM 1010 (80 KeV), after applying 10 mL of the aqueous powder dispersions to a copper grid, and by Scanning Electron Microscopy (SEM) (JEOL JSM-7200F Field Emission Scanning Electron Microscope at an accelerating voltage of 10 kV). Prior to SEM observation, the samples were sputter-coated with a thin palladium/platinum layer (Cressintong 208HR). The printed composites were also analyzed by SEM. In this last case, the specimens were previously broken under cryogenic conditions. Confocal microscopy was used to determine the surface roughness of the conductive printed composites with pTSA-PANI, by using a PLu 2300 Sensofar^®^ optical imaging profiler. Images were captured using an EPI 10×-N objective, a depth resolution of 2 μm, and a lateral resolution of 1 nm. Roughness parameters such as R_a_ (average roughness), R_v_ (average maximum valley depth), and R_p_ (average maximum peak height) were obtained using SensoMaP 5.0.4 software. At least five measurements were performed for each sample in order to calculate the average values and standard deviations.

For lignin and pTSA-PANI powders, the FTIR spectra were performed in Potassium Bromide (KBr) pellets between 4000 and 400 cm^−1^. The post-cured printed films were analyzed in ATR mode between 4000 and 550 cm^−1^ with a 4 cm^−1^ resolution over 64 scans. The degree of the acrylate double bond conversion (DBC%) based on Equation (2) was also calculated. For each sample, the average spectra of three replicates were examined.

Thermogravimetric Analysis (TGA) (TGA 4000—Perkin–Elmer, Waltham, MA, USA) of the cured films was performed under a nitrogen atmosphere at 50 mL min^−1^ using ceramic crucibles (60 μL) as composite holders. The heating rate was 10.0 ± 0.1 °C min^−1^ from 50 up to 700 °C.

Tensile stress–strain mechanical properties were characterized using an Instron 5569 universal testing machine (Instron Canton, Norwood, MA, USA). The analysis was performed using a cross-head speed of 5 mm min^−1^ until failure, at room temperature. At least five dog-bone-shaped specimens were tested following ISO 527 (dimensions 75 × 13 × 2 mm; width of narrow section 5 mm). Measurement of the hardness of the composites with a Shore “A” Durometer (Durotech M202) was carried out on the dog-bone-shaped specimens at a distance of ~6 mm from the edge of the material after 15 s of force application. The measurements were taken at 10 measuring points on each sample and the mean values and standard deviations were calculated according to ISO 868:2003 [33].

Surface wetting measurements were carried out with a Theta Lite Attention tensometer (Biolin Scientific, Gothenburg, Sweden) and the software program “One Attention”. The static water contact angle (θ) formed by a single droplet was measured at least five times on dry samples using a 4 μL sessile drop of deionized water as test fluid at room temperature and the average values are reported. Images were recorded every 10 s. 

The electrical conductivity (*σ*) of the samples was calculated from the electrical resistance data by the four-probe method (LORESTA-GP, Mitsubishi Chemical, MCP-T610, Tokyo, Japan) at room temperature. For pTSA-PANI, square compression molded pellets of 2.5 cm × 2.5 cm × 0.5 mm were employed. The values reported are the mean of at least eight readings measured on three different samples.

## 3. Results

### 3.1. Prepolymerization Studies

Prior to 3D printing, the effect of lignin on viscosity, UV-spectroscopy, photopolymerization kinetics and the Jacob’s working curves were evaluated, being these parameters significantly interesting for a DLP process and the quality of printed parts.

#### 3.1.1. Viscosity

Viscosity is one of the parameters to be taken into account to achieve satisfactory prints. In general, low viscosities are desired to allow the proper coating of the last printed layer or the surface of the immersive platform [34]. On the contrary, too high a viscosity means longer exposure times, as well as limited adhesion of layers to each other and to the printer platform [28,31].

The viscosity values of all liquid formulations as a function of shear rate and Lignin amount at room temperature are depicted in Appendix A, whereas the viscosities at 1 s^−1^ are compiled in Table 1. It is obvious that the higher the lignin content, the greater the viscosity of the formulations at 1 s^−1^, since the presence of lignin limits the mobility of the polymer chains [25,35]. Increased lignin content (0–4 wt. %) raised the viscosity of the polymer from 0.011 Pa s to 0.072 Pa s at 1 s^−1^. Although the reported viscosities for current commercial resins are usually higher, in the range of 0.85−4.5 Pa [36], the viscosities of all the lignin formulations remain low enough to promote the resin layer uniformity in the 3D printer in use and are comparable to the viscosities of PANI, PANI-Lignin and PANI-MWCNT formulated with the same base acrylic resin and 3D printed successfully [31,32].

In contrast to the base formulation (LG0), which has Newtonian behavior in the range between 1 s^−1^ and 10^3^ s^−1^, all doped formulations show typical shear thinning behavior at 1 s^−1^ (Appendix A and Table 1), in agreement with previous studies on resins incorporating nanofillers [30,35,37]. This shear thinning behavior can facilitate the spreading of homogeneous layers in vat polymerization techniques [38]. Thus, from the viscosities point of view, the studied formulations are viable for creating any type of design by 3D printing.

#### 3.1.2. UV-Visible and Real Time-FTIR Spectroscopy

The UV–Vis absorption spectra between 330–800 nm of lignin is shown in Figure 2A. Lignin has absorption bands around 280–300 nm related to the phenolic hydroxyl and aromatic moieties [39]. The absorptivity value at 405 nm, *ε*, has been calculated from the slope of the regression line obtained when plotted, the experimental absorbance values, *A*, vs the concentrations of the filler in dispersion, using the Lambert–Beer law (Equation (1))
(1)A=ε×b×c
where *b* is the length of the UV pathway (1 cm for the cuvettes used) and *c* is the concentrations of the tested fillers. The calculations have been performed on the basis of filler suspensions, thus the Lambert–Beer law is an approximation since no light scattering effects have been taken into account. The wavelength 405 nm selected is the critical wavelength on which the UV lamp of the printer has its highest radiation power. The calculated *ε* value is 1.45 g. L^−1^. cm^−1^ (r^2^ = 0.9967). This outcome suggests that UV absorption by lignin competes with the photoinitiator and hinders the curing process of the dispersions.

The influence of lignin (0, 0.5, 1, 2, 3, and 4 wt.%) on the photocuring rate of the acrylate resin has been studied by in situ ATR-FTIR spectroscopy. The bands related with the C=C double bond of the acrylate groups gradually disappeared, specifically 1636 cm^−1^ (ν_C=C_, doublet), 1409 cm^−1^ (in plane deformation, scissoring, δ_=CH2_), 984 cm^−1^ and 810 cm^−1^ (out of plane deformation, δ_=CH2_) (indicated by arrows in Figure 2B). To follow the evolution of the polymerization reaction, the degree of double bond conversion (DBC%) was calculated according to Equation (2) [40]. The decrease of the band at 810 cm^−1^ was normalized to the carbonyl ester stretching band (ν_C=O_) of the resin at 1728 cm^−1^, using this last one as the internal reference; *t* indicates the time irradiation [40]:(2)$$\mathrm{DBC}\%=\frac{{\left(A_{810}/A_{1728}\right)}_{t=0}-{\left(A_{810}/A_{1728}\right)}_{t}}{{\left(A_{810}/A_{1728}\right)}_{t=0}}\times 100\%$$

Figure 2C shows the conversion curves of DBC% versus irradiation time (s), were fitted with the Boltzmann sigmoidal model [41]. Three sections can be observed, corresponding to induction, propagation, and equilibrium or ultimate degree of conversion (DBC_∞_%). The maximum rate of polymerization (*R_P_*) has been calculated by Equation (3) for the conversion interval between 25–55% [32,40].
(3)$$R_{P}=[M_{0}]\frac{{\left(A_{810}\right)}_{t1}-{\left(A_{810}\right)}_{t2}}{t_{2}-t_{1}}$$

Being [$M_{0}$] the initial concentration of acrylate double bonds (5.63 mol l^−1^ considering that HDODA is a bi-functional monomer and EGPEA is monofunctional), ${\left(A_{810}\right)}_{t1}$ and ${\left(A_{810}\right)}_{t2}$ are the areas of the at the irradiation times *t*_1_ and *t*_2_, corresponding to 25 and 55% conversion, respectively [32].

The maximum rate of polymerization $(R_{P}),$ the induction period and ultimate degree of conversion are shown in Table 1. Longer retardation in polymerization initiation and slightly lower final conversions are seen for lignin contents ≥1 wt.%. Most importantly, as shown in Figure 2C inset, there is a pronounced exponential decrease in the rate of photopolymerization (r^2^ = 0.997) when increasing the filler amount from 0.5 wt.% onwards. This is an expected behavior, partly due to the absorption of UV light, which competes with the light adsorption of the photoinitiator. The role of lignin UV blocker is due to the presence of UV-active functional groups such as C=O and aromatic rings [7,25]. The decrease in curing kinetics observed for lignin composites could also be related to the reported role of the phenolic groups of lignin as free radical scavengers and antioxidants [42,43,44]. On the contrary, some types of modified lignin have proved to have a certain photoinitiator ability when adequately combined with an amine co-initiator [39].

#### 3.1.3. Jacobs Working Curves

In this section, the printability of the developed formulations by DLP is evaluated. Owing to the UV absorption of lignin and its reported effect as a free radical scavenger, the photopolymerization reaction is hindered, jeopardizing successful printing. To determine the optimal exposure times for each formulation, Jacobs working curves for LG1, LG2 and LG3, as well as printing tests, were carried out. Jacob working curves were calculated according to Equation (4) [45], where *C_d_* is the cured depth (µm), *D_p_* is the penetration depth (µm) of the light into the resin, *E*_*max*_ is the light irradiation dosage on the surface (mJ cm^−2^) and *E_c_* is the critical exposure required for polymerization (mJ cm^−2^) [32]. As previously indicated, the light intensity of the 3D printer was 9 mW.cm^−2^ and the layer thickness (z) was 25 µm for all samples.
(4)$$C_{d}=D_{p}\cdot \ln\left(\frac{E_{max}}{E_{c}}\right)$$

As shown in Figure 2D, the correlation coefficients (R^2^) of the logarithmic regression lines were all higher than 0.99. Based on Equation (4), the photosensitive parameters related with the intrinsic properties of the resin, *E_c_* and *D_p_*, were calculated. For LG1, LG2 and LG3, the *E_c_* values were 11 mJ.cm^−2^, 13 mJ.cm^−2^ and 21 mJ.cm^−2^ respectively. Regarding the penetration depth of the resin, the *D_p_* values were 58 μm, 41 μm and 40 μm. As expected, the general trend is that as the amount of lignin increases, the amount of energy needed to induce polymerization (*E_c_*) increases, while penetration depth (*D_p_*) decreases.

For lignin contents ≤2 wt.% the critical exposure (*E_c_*) to induce polymerization remains actually unchanged, whereas it doubles its value for 3 wt.% lignin. Notwithstanding, the *E_c_* values are in the same range or lower compared to those reported for commercial nonconductive resins [25,28,36]. The *D_p_* of LG1 (58 µm) lies within the low range of commercial resins without conductive fillers tested by Bennet et al. (53–568 µm), whereas *D_p_* values decreased around 31% for 2 and 3 wt.% lignin contents [36]. Low *D_p_* values have the advantage of allowing accurate control of the polymerization process and minimal over-cure, although the shortcoming of longer building times [36]. Further, the curing parameters are similar to those calculated for PANI and PANI MWCNT composites fabricated with the same acrylic matrix and photoinitiator content [23,31,32]. 

To ensure that a resin is suitable for a specific printer, it is advisable to calculate the *C_d_* corresponding to the light source employed from Equation (4) and compare it with the layer thickness (z) [46]. The layer thickness must be equal to or less than the curing depth. The exposure times indicated in Table 1 were set based on *D_p_* and *E_c_* values and by test prints; their corresponding *E_max_* were calculated considering the printer energy used (9 mW.cm^−2^). The *C_d_* values for LG1, LG3, and LG3 were 81, 51, 30 μm, respectively. To ensure good adhesion between layers, the target is *C_d_* > z, since the stiffness of a polymer below the gel point would hinder the printing process [31]. In this way, the obtained values for LG1 and LG2 (*C_d_* = 81 and 51 µm, respectively; z = 25 µm) show that these formulations are suitable for 3D printing, whereas the printing of LG3 (*C_d_* = 30 µm) has proved difficult and less reproducible.

### 3.2. Composites Characterization

#### 3.2.1. Structural Characterization: Morphology and ATR

TEM and SEM imaging were performed to examine the morphology of lignin (Appendix A). TEM and SEM images portray an irregular distribution of particles with a wide range of sizes that consist on aggregates of individual spheres. Appendix A shows an isolated sphere of about 200 nm in diameter. These spherical particles are probably formed of disordered entangled chains that have shrunk into a “collapsed ball” [21].

The morphological differences between the 3D printed films of pure acrylate resin and the lignin compounds were further evaluated by SEM. Analysis of the composites films surfaces (Figure 3) showed polymer-rich surfaces as lignin is not observed in the images, indicating that the filler is embedded within the resin. Moreover, the surface of the composite films became smoother upon increasing the lignin loading, with fewer and shallower scratches visible on the surface, which points to an increase in surface hardness. The observation of a reduced surface roughness of polymer nanocomposites films with increased lignin content has been previously reported [15,47]. 

Concerning the cryo-fractured cross-section shown in Figure 4, all the samples showed the layer stack gaps every ~25 μm, consistent with the layer thickness set at 0.025 mm. Nevertheless, there are some differences related with the fracture mechanism, the crack formation, and propagation. The LG0 sample shows a much smoother cross-section with several long cracks in accordance with the rubbery nature of the polymer (Figure 4); opposite to this, the rigid lignin microparticles augment the number of stress concentration spots, leading to an increasingly uneven and rougher fracture surface for composites with lignin contents ≥ 2%, as previously reported [26]. This behavior is enhanced by both the reduction on the UV-photopolymerization rate and the lignin aggregation and uneven dispersion in the matrix, which leads to increasing gaps and holes (Figure 4C,D insets). A poor dispersion of the filler will provide more concentrated stress locally, negatively affecting the mechanical properties of the material [27].

The KBr FT-IR spectrum of lignin is plotted in Figure 2B in comparison with the ATR spectra of the liquid monomers mixture, the pristine acrylic matrix, and the composite LG35. The spectrum of lignin showed typical lignin patterns after organosolv fractionation. The stretching vibrations of hydroxyl, CH alkane, carbonyl, and typical aromatic skeletal vibrations were observed at 3425 (ν_OH_), 2936, and 2851 (ν_C-H_), 1734 (ν_C=O_), 1616, 1508, and 1425 (ν_C-=C_) cm^−1^, respectively. An absorption at 1463 cm^−1^ related to C-H bending vibration (–CH_2_, –CH_3_) can be seen. Furthermore, the bands at 1239 and 1044 cm^−1^ may be ascribed to the C=O bending and aromatic CH in plane deformation of the guaiacyl (G) unit, whereas those centered at 1328 and 1117 cm^−1^ have been assigned to the same functional groups of the syringyl (S) unit. [16,48,49].

In relation with the pristine acrylic resin, the strongest bands in the spectrum centered at 1728 and 1156 cm^−1^, are due to the stretching vibrations of the carbonyl bond (ν_C=O_) and the asymmetric stretching of the C–O–C bond (ν_C–O–C_), respectively [23]. The presence of lignin is not detected in the spectra of composites with lignin contents below 3 wt.% due to the overlapping of most lignin signals with the bands of the acrylic matrix. Nonetheless, for higher lignin contents, as in the spectrum of LG35 (Figure 2B), the filler can be perceived by a slight increase in the absorbance region between 3600–3200 cm^−1^. Unlike previous reports, which used higher amounts of unmodified lignin (5 and 10 wt.%) [44], no bands shifts, or new bands, corresponding to the H-bonding between the acrylic matrix and lignin, have been detected in the ATR spectra, which is coherent with the observations made by SEM, suggesting that lignin is mostly embedded within the resin.

The effects of lignin on the degree of monomer conversion (%DBC) of the post-cured printed films were confirmed by ATR-FTIR. The almost complete disappearance of all the bands allotted to the C=C double bond of the acrylate groups revealed high degrees of monomer conversion on both sides of the films, irrespective of lignin loading. Despite the observed effect on the rate of polymerization, no differences with respect to pristine acrylic matrix have been found within experimental error (DBC% = 97.5 ± 0.6 and 96.6 ± 0.8 for LG0 and LG35, respectively). As discussed by previous authors, these results proved the need and efficiency of the UV post-curing step in completing the photoreaction [26]. 

#### 3.2.2. Thermal and Mechanical Properties of Printed Composites

The thermal and mechanical behavior of lignin-containing materials is key to ensuring their processability and good final properties. Lignin can be used in polymers as a stiff filler with a purpose similar to the role played in plants providing cell wall rigidity. However, the reinforcing effect of lignin is highly influenced by several factors, such as the lignin source, content, polymer resin, and the printing techniques used [26] that can even cause undesired effects. In this way, lignin can worsen the mechanical properties of the polymer due to its naturally variable composition and irregular structure or introduce heat instability due to its phenolic moieties [44]. For these reasons, it is important to assess the final properties of the obtained lignin composites. At first, the thermal stability of lignin, pristine acrylic, and the printed composites were evaluated by TGA (Figure 5A and Table 2). 

Lignin shows three-step weight-loss behavior. The first loss step below 140 °C is attributed to the release of volatile components (2.5 ± 0.2%). Between 146 and 500 °C lignin loses 55.8 ± 0.5% of the initial mass in two thermal degradation processes with maximum degradation temperatures (T_max_) at 223 and 346 °C. The degradation below 400 °C is mainly caused by the fragmentation of the weak inter-unit linkages (β-O-4) [50,51,52]. The loss of mass above 500 °C can be explained by the decomposition of aromatic rings and the cracking of C–C linkages [50,51,53], followed by the release of OCH_3_ groups from aromatic rings [50,54]. After 700 °C, there was still a 35.1 ± 0.5% mass attributed to the formation of highly condensed aromatic structures [51,54]. 

Regarding pure acrylic resin, it shows two-step degradation behavior. The first small loss step is attributed to the release of volatiles (T_max_ = 188 ± 2.8. °C, 2.6 wt.%). The scission of the main polymeric backbone chain occurs within 350–450 °C [30,55], leaving a very small residue at 700 °C; therefore, the copolymer can be considered thermally stable up to 350 °C. 

Concerning 3D printed lignin composites, they all exhibit degradation profiles similar to that of the pristine acrylic resin. No variation on volatiles evaporation is detected, within experimental error, confirming the high degrees of conversion observed by ATR and the efficiency of the post-curing step. Nevertheless, upon increasing the amount of lignin from 0.5 to 4 wt.% T_10_ and T_onset_ linearly decrease by about 6 and 4%, respectively (r^2^ = 0.98 and 0.97). By contrast, although the maximum rate of weight loss temperature (DTG) shifted from 426.5 °C for the neat acrylic resin to 420.5 °C for lignin contents of 0.5% wt.%, the addition of higher amounts of lignin does not lead to further changes. Finally, a small increase in weight residue is observed upon increasing lignin content. The small changes in thermal behavior are coherent with the lower thermal stability of the lignin compared with the acrylic matrix [56] and with the absence of interactions between the matrix and lignin. At any rate, all composites are thermally stable up to 300 °C. 

Lignin has been investigated as reinforcement in the field of 3D printing to enhance the mechanical properties of the printed materials, although the reinforcing effect of lignin greatly differs depending on the lignin source, content, polymer resin, the printing techniques, and the post-curing step [26,28,56], as previously indicated. To evaluate the mechanical properties of the composites, surface hardness (Shore A) and uniaxial tensile tests till rupture were performed according to ISO 868 and 527, respectively. The stress–strain curves are plotted in Figure 5B and the calculated modulus, stress at break, and elongation at break are shown in Table 3. 

The addition of low values of lignin does not modify the hardness of the acrylic resin, whereas a small increase (~5%) is detected for composites with lignin contents ≥3 wt.%. This small effect of unmodified lignin on the hardness of rubbery resins can be expected due to the lignin stiffness, which is higher than that of the rubbery matrix and the lack of interaction between lignin and the resin [57].

No variations in tensile parameters are detected with respect to LG0 samples at low lignin values (≤2 wt.%) within experimental error. For the tensile modulus results, a similar trend as hardness outcomes is appreciated. The small increase in tensile modulus for lignin contents ≥ 3 wt.% has been attributed by some authors to the rigid phenolic units in lignin [26]. At the same time, a reduction around 25–28% on the elongation is revealed. This diminution is explained by the stiffness increase together with the lack of strong interaction between lignin and the matrix. The stress is not transmitted from the polymer matrix to the filler and vice versa, so the material breaks more easily [27,57]. For the tensile strength, although the variations detected for lignin values of 3.5 and 4% fall within experimental error, the observed trend is in agreement with previous publications explaining that reduction at higher loadings may be due to imperfect curing caused by lignin aggregation and uneven dispersion within the acrylic matrix, producing gaps and holes in the composites [56], which can be perceived in the SEM pictures of Figure 4D inset.

#### 3.2.3. Contact Angles (CA)

There are relatively few publications regarding the wettability of additively manufactured materials. Depending on the production process as well as on the chemical composition, specifically the effect of additives, materials can differ in their surface morphology and consequently in their CAs [58]. The influence of lignin on the average water contact angles of printed composites is displayed in Figure 6. The contact angle for the control sample (86.6° ± 0.6) showed a rather hydrophobic nature, which agrees with that reported in the literature for acrylic rubbers with contact angles ranging between 78–89° [59,60]; the contact angle decreases slightly for lignin contents of 0.5 wt.%. and stabilizes at 70° ± 2 for lignin contents ≥ 1 wt.%. The reason may be dual. On the one hand, lignin possesses polar and hydrophilic groups (carboxyl, phenols…), therefore, more hydrophilic groups are distributed on the surface, which reduce the interfacial tension and increase its hydrophilic character, as has been reported for different polymer matrices [61] of hydrophobic nature. On the other hand, surface roughness slightly decreases with lignin content, as shown in Figure 3. Opposite to this, lignin confers a hydrophobic character on composites based on more hydrophilic resins like polylactic (PLA) or polyhydrobutyrate (PHB [62]), indicating that surface properties can be widely tuned with lignin.

This decrease in contact angle with small lignin contents may contribute to improving the impression because the resin wets the surface of the platform and the anterior layer better. Further, it facilitates subsequent painting, if necessary, for finishing the part. It has been postulated that adequate surface wettability can aid in the production of 3D printed elements for applications involving interactions with fluids, such as antistatic coatings, electrochemical sensors, microfluidic devices, among others [60].

### 3.3. Use of Lignin as Dispersant of Conductive Fillers 

With the aim of evaluating the potential application of lignin for the manufacture of functional materials, the filler dispersing effect of lignin in conductive matrices has been evaluated in this section. For this purpose, two photocurable acrylic formulations with 5 wt.% conductive filler pTSA-PANI have been developed. To one of these formulations, 1 wt.% lignin has been added (LG1PANI5) to assess the potential improvement of its properties compared with those of the lignin-free reference (LG0PANI5).

#### 3.3.1. pTSA-PANI Characterization

At first, the properties of pTSA-PANI were characterized by TEM, SEM, UV-vis spectroscopy, FTIR, elemental analysis and electrical conductivity. Regarding the morphology, neat pTSA-PANI consists of long fibrillar chains (≈ 0.03–0.05 × 1.0–1.5 μm) partially surrounded by short nanosized granular structures with a great tendency to aggregate (Appendix A). The aggregation of the fibrillar and granular structures leads to the formation of flat platelets interwoven by individual fibrillar chains (Appendix A). The hollow-tube morphology described by Khalid et al. for PANI-PTSA is not perceived [63]. The key factor that determines the pTSA-PANI structure is the acid dopant:aniline ratio. As a general rule, the fibrous structure of pTSA-PANI, favorable for high conductivity, has been described for high acid dopant: aniline ratios, as is the present case; by contrast, lower acid dopant concentration leads to coral-like structures. Nevertheless, when the volume of the acid anion is large, as is the case for pTSA, the slower movement of molecules decreases the doping rate, which is responsible for the decrease of pTSA-PANI fiber structure [64] and the observed mixed morphology.

Both the UV-visible (Figure 2A) and FTIR spectra (Figure 7) confirm that the synthesized pTSA-PANI is in the protonated doped state. The UV-Vis spectra of pTSA-PANI dispersed in glycerin shows a band between 330–800 nm centered at 450 nm and the upward slope of a band located above 800 nm. These features are in accordance with previous literature that describes for pTSA-PANI 2 peaks in the same range at 433 and 800 nm, assigned to the shift from polaron to π* band and from π to polaron band of the doped pTSA-PANI chains, respectively [65,66]. By contrast, Beygisangchin et al. reported the maxima of these bands at neatly lower wavelengths, 321 and 578 nm, respectively [66]. The absorptivity coefficient at 405 nm was calculated in a similar way to lignin, as 1.03 g L^−1^ cm^−1^ (r^2^ = 0.9656). It was slightly lower than the value obtained for lignin, so a similar hindering of the curing process of the resin is expected.

All the characteristic peaks of -PANI plus those of the dopant, p-Toluensulfonic acid, are observed in the FTIR spectrum of d pTSA-PANI (described in detail in the Appendix A) [63,67]. The oxidation state of the polymer was calculated through the intensity ratio of the Q/B absorption bands (I_Q_/I_B_) at 1561 cm^−1^ (Quinoid rings, Q) and 1467 cm^−1^ (Benzenoid rings, B); the emeraldine type structure corresponds to a value of 1.0. Oxidation mostly depends on the oxidant concentration and the pH of the reaction medium [68]. The I_Q_/I_B_ ratio, 0.82 ± 0.03, is somewhat lower but still close to unity and similar to that obtained for PANI-DBSA synthesized under the same conditions [29]. The intrinsic oxidation of pTSA-PANI is an important feature as the quinoid imines are preferentially protonated in the protonic acid doping. The level of doping when sulfonic acids are employed can be calculated from the elemental analysis. Hence, the S/N bulk atomic ratio derived from the elemental analysis data, 0.50, agrees with a high doping level [69,70].

The electrical conductivity of pTSA-PANI filler was on average 7.6 ± 0.5 S. cm^−1^, which is in the range of good semiconductors and is a result of the fibrous morphology, the high doping level, and the oxidation state. This conductivity is similar to the values of PANI-HCl previously obtained by this research group [31] and lies within the range of several reported pTSA-PANIs synthesized by oxidative polymerization with conductivities spanning from 1.46 to 34.8 S. cm^−1^ [63,66,71,72]; by contrast, the electrical conductivity is 2 orders of magnitude higher than pTSA-PANI obtained by redoping [73].

#### 3.3.2. Characterization of Printed pTSA-PANI Composites

Figure 8A,B depicts the physical appearance of the two pTSA-PANI printed films. As evident from the photographs, LG1PANI5 shows a homogenous shiny surface in contrast to the uneven matte surface with pTSA-PANI lumps protruding from the lignin-free sample. The smoother morphology and more uniform distribution of pTSA-PANI aggregates in the presence of 1 wt.%. lignin is confirmed by confocal microscopy. As can be appreciated in Figure 8C,D, the relatively small pTSA-PANI clusters, evenly distributed on the surface of L1PANI5, contrast with the presence of large lumps in the lignin-free film; as a result, all the roughness parameters are clearly lower for the LG1PANI5 sample (Table 4). Its average surface roughness (R_a_) decreases by 61% compared to that of LG0PANI5 film, suggesting a clear improvement in the dispersion of the fillers by incorporating lignin. Regarding the R_v_ and R_p_ parameters, a decrease of 60% is observed in R_v_ of LG1PANI5 compared with LG0PANI5, while for R_p_, this decrease is 29%. This parameter presents a relatively high standard deviation which can be associated with the presence of scratches, cracks and irregularities unevenly distributed across and throughout the surface; the scratches are due to low surface hardness and the pTSA-PANI clusters to poor dispersion of the conductive filler within the acrylic matrix. These results agree with those reported by Yang et al. [24] concerning the surface morphologies of different PANI-lignin films, spin-coated on ITO substrates. They observed by AFM smoother and more uniform surfaces in comparison with pure PANI. These smoother surfaces were attributed to the better dispersibility [24].

R_a_ value indicates the absolute values of the profile heights over the evaluation length, R_p_ the height of the main peak structures. and R_v_ estimates the average depths of the fissures.

To further monitor the dispersion condition of PANI in the acrylic matrix, the morphologies LG0PANI and LG1PANI are assessed by SEM. The surface morphologies are consistent with the physical appearances and images were taken by confocal microscopy. In relation with the cryo-fractured cross-section of both films (Appendix A), the horizontal lines corresponding to the individual printing layers are perceived together with phase separation, as great PANI agglomerates lodged in cavities, completely detached from the matrix, are observed. These images prove the little interaction between the acrylic matrix and the conductive charge in the absence of lignin, whereas the addition of lignin improves the embedding of PANI in the matrix, as shown in more detail in Appendix A. Further, there are also some differences related to the fraction mechanism, as the crack’s formation and propagation are greater in LG1PANI5 with respect LG0PANI5 or even LG2 and similar to LG3.

From another point of view, the contact angles were also determined in order to deepen the knowledge about surface properties and their ability to create new interfaces (Figure 6), LG0PTSA 5 and LG1PTSA5 3D printed film’s contact angles were (63.90° ± 2.64) and (61.70° ± 2.90), respectively. No differences are found between them within experimental error (Student test, *p* > 0.05), although both surfaces are more hydrophilic than the neat acrylic resin (86.60° ± 0.60) and LG1 (69.92° ± 2.00). The decrease in contact angles compared to the neat resin is due to the more hydrophilic nature of PANI salts and the presence of amine groups in its structure, although the contact angles ultimately depend on the dopant used [74]. Contact angles of 69.9° ± 1.7 have been reported for polymer membranes coated with pTSA-PANI [75]. The higher hydrophilicity provided by PANI compared to lignin explains why no differences are observed between the two PANI composites. Further, the difference in surface roughness between LG0PTSA5 AND LG1PTSA5 is not reflected in the contact angles, probably due to the relatively high experimental error in the measurement. At any rate, these outcomes support the good wetting properties of both composites.

Regarding the ATR spectra of the composites, L0PANI5 and L1PANIT 5, in Figure 7 the specific bands of PANI and lignin are not clearly identified on the printed surface, as the spectra of both composites are practically superimposable to that of the pure resin, indicating that fillers are mostly embedded within the resin. Nevertheless, a detailed observation allows perceiving minor increases in the absorbance region between 3400–3200 cm^−1^, corresponding to the stretching vibrations of the NH groups of PANI and the OH groups of lignin and a small shoulder at around 1688 cm^−1^, which may be related to the stretching of the C=O groups of the acrylic resin H-bonded with PANI. Further, a small band is distinguished at 1640 cm^−1^ (C=C stretching region of aromatic rings present in the acid dopant (pTSA)). These changes are more noticeable in the composite without lignin (L0PANI5), where an additional band at 1540 cm^−1^ (Q band of PANI) is observed.

The influence of 5 wt.% pTSA-PANI without o with 1 wt.% lignin on the degree of monomer conversion (DBC_∞_%) of the post-cured printed films was studied. The DBC_∞_% values of the composites showed no differences between them, within experimental error, but were ~3–4% lower than the DBC_∞_% of the pristine resin (DBC%: 97.5 ± 0.6, 93.8 ± 4.8 and 94.3 ± 2.8 for LG0, L0PANI5 and L1PANI5, respectively). These are expected results similar to those obtained in previous works for composites of the same acrylic matrix [23,31] filled with similar contents of PANI-HCl, and are explained by the relatively high absorptivity value of pTSA-PANI at 405 nm, which strongly competes with the light adsorption of the photoinitiator. Besides, the standard deviations of these values are greater in the composites than in the reference sample due to poor filler dispersion and is worse in the composite without 1wt.% lignin. This outcome agrees with the results of confocal microscopy showing the better inclusion of PANI aggregates within the resin in LG1PANI5.

Last but not least, the electrical conductivity of the neat acrylic resin is lower than 1.0 × 10^−8^ S cm^−1^, which is the limit of detection of the instrument. The conductivity increased with the inclusion of 5 wt.% pTSA-PANI up to (1.7 ± 0.1) × 10^−7^ S cm^−1^. When 1 wt.% Lignin is added to the formulation with 5 wt.% pTSA-PANI (PANI5LG1), the conductivity increases by an order of magnitude to (1.6 ± 0.1) × 10^−6^ S cm^−1^, suggesting a small enhancement in the electrical properties of the ternary composites. These results agree with confocal microscopy images, since the presence of lignin suggests an improvement of the filler dispersion in the matrix, which translates into the existence of a continuous network of pTSA-PANI across the insulating acrylic matrix. Henceforth, these values are in the range of semiconductors. Similar conductivity on the order of 10^−6^ S cm^−1^ was previously achieved by conductive composites based on photocurable epoxy resins doped with 15 wt.% pTSA-PANI [76,77]. Regarding the incorporation of PANI as conductive filler in photocurable resins, very few studies exist, and the results obtained show medium–low conductivity values. Table 5 summarizes the main recent research on light-curable 3D printed resins with electrical properties based on PANI. Although PANI/MWCNT has been extensively studied in the field of electronics, only one previous research used this filler in a photocurable resin for SLA/DLP.

In contrast, as far as the authors are concerned, there are no data on electrical conductivity of DLP 3D printed composites using pTSA-PANI filler, nor any study that employs unmodified lignin as a compatibilizer. Only a few works have been found in the literature about 3D-DLP printing of conductive light-curing acrylic resins based on polyanilines with other dopants and other conductive fillers. The electrical conductivity reached in the current study is higher than those reported for photocurable acrylic and polyurethane 3D-DLP printed composites loaded with HCl doped PANI between 1–6 wt.% (10^−10^–10^−7^ S cm^−1^) [78,79]. The data are slightly lower than those obtained for DLP 3D printed acrylic resins filled with 0.3–0.6 wt.% carbon nanotubes (10^−4^–10^−5^ S cm^−1^) [80,81,82] and similar contents of PANI-HCl (10^−5^ S cm^−1^) [31] or a mixture of PANI/MWCNT (10^−4^ S cm^−1^) [32]. Simultaneously, the conductivity values achieved are similar or higher than those reported for printable photocurable resins using 16 wt.% (10^−6^ S cm^−1^) [83] and 1 wt.% (10^−11^ S cm^−1^) [84] of Ag nanoparticles or 2 wt.% reduced graphene oxide (rGO) (10^−7^ S cm^−1^) [85]. By contrast, some authors have reported conductivity values of the order of 10^−2^–10^−3^ S cm^−1^ with 6 wt.% of rGO [86] or using 20 wt.% of PEDOT [60]. Nevertheless, the use of high amounts of filler is not desirable considering their cost and the negative effect on the printing process of the material [86].

## 4. Conclusions

This work reports on the systematic study of the structure–property relationship of 3D printed composites with increasing amounts of unmodified organosolv lignin in a light-curable acrylic matrix. Amounts of lignin below 3 wt.% can be successfully and reproducibly printed by LCD vat polymerization. This limit is due to the strong UV absorption at the critical wavelength of the UV lamp of the printer, which greatly decreases the rate of polymerization and the cure depth. The printed polymer composites have high degrees of conversion, rich acrylic surfaces, hardly any interactions between the filler and the matrix, and discrete lignin aggregates within the polymer matrix. For composites with lignin proportions below 3 wt.%, the bulk properties of the pristine resin are maintained, simultaneously improving the surface properties, resulting in smoother surfaces, increased Shore A hardness, and better wettability. These properties are beneficial from the point of view of 3D printing elements for applications involving interactions with fluids and facilitate subsequent finishing of the parts if necessary.

The effect of organosolv lignin as a dispersant of pTSA-PANI was assessed. A clear improvement in the dispersion of the conductive filler was achieved with as little as 1 wt.% lignin in the formulation, resulting in more homogeneous samples with less surface roughness, better appearance, and electrical conductivity enhanced by one order of magnitude, up to 10^−6^ S cm^−1^, without negatively affecting printability.

In brief, the present research opens the possibility of developing a range of novel solvent-free, eco-friendly photocurable nanocomposites for the fabrication of functional materials at low costs, valorizing a natural resource such as lignin.

## Figures and Tables

**Figure 1 polymers-14-04164-f001:**
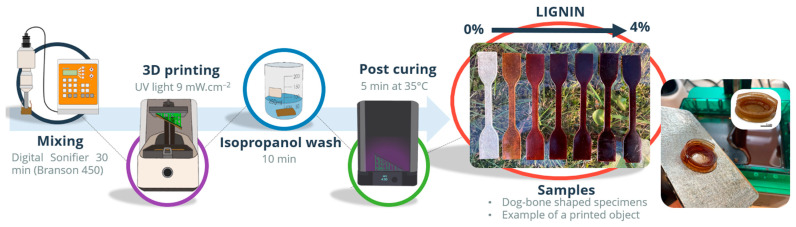
Diagram of sample preparation.

**Figure 2 polymers-14-04164-f002:**
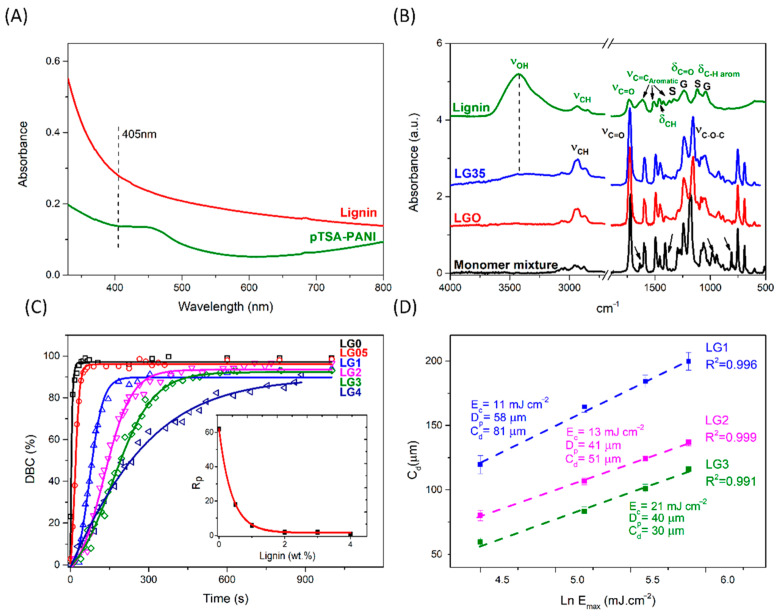
(**A**) UV-visible absorption spectra of lignin and pTSA-PANI in glycerin at 150 ppm. (**B**) ATR-FTIR spectra of the monomer mixture, pristine acrylic (LG0), LG35, and Lignin. (**C**) ATR conversion curves versus irradiation time for acrylic dispersions with 0.5, 1, 2, 3, and 4 wt.% lignin. The experimental data (symbols) were fitted with Boltzmann sigmoidal equation (lines) and (**D**) Jacobs working curves showing cure thickness as a function of the natural log of UV dosage for LG1, LG2, and LG3 formulations. Linear regressions are depicted.

**Figure 3 polymers-14-04164-f003:**
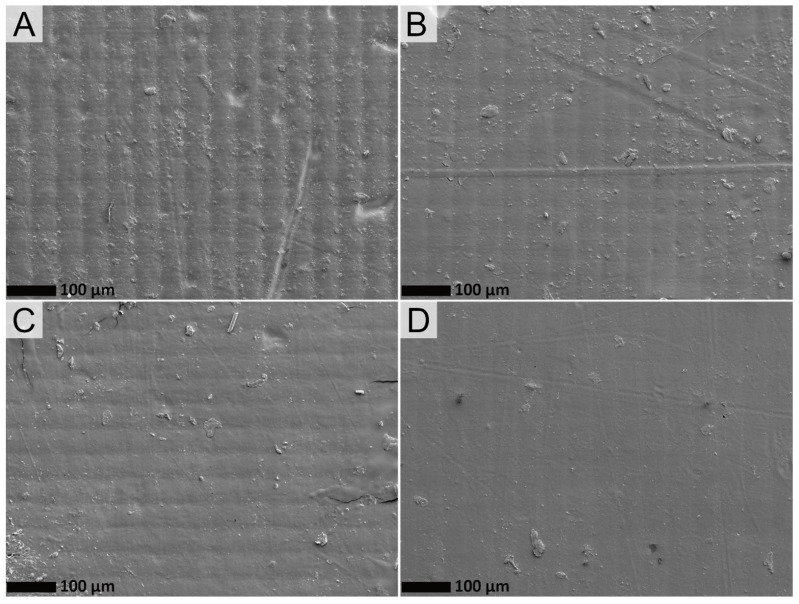
Representative SEM images of the films surfaces of (**A**) LG0, (**B**) LG1, (**C**) LG2, and (**D**) LG35 with a magnitude amplification of 500×.

**Figure 4 polymers-14-04164-f004:**
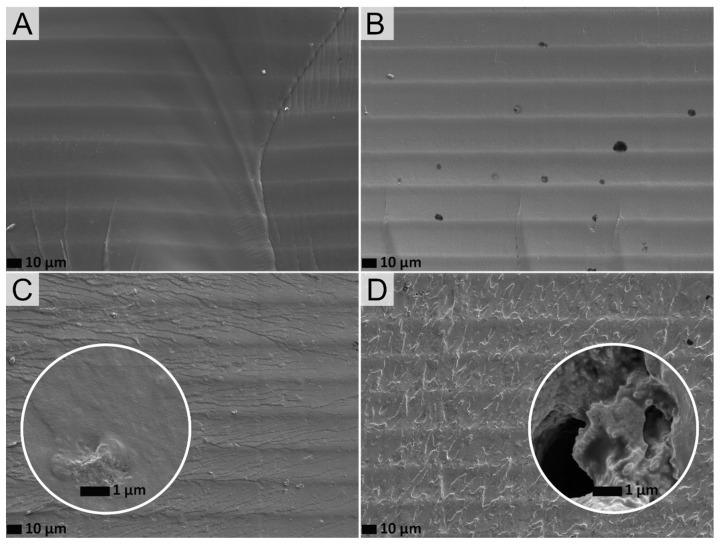
Representative SEM images of cryo-fractured cross-sections of (**A**) LG0, (**B**) LG1, (**C**) LG2, and (**D**) LG35 with magnitude amplification of 500× and 10,000× for the inset.

**Figure 5 polymers-14-04164-f005:**
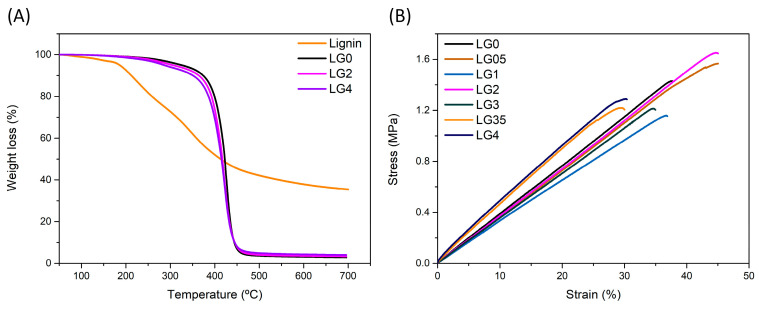
(**A**) TGA curves of Lignin and printed composites LG0, LG2, and (**B**) Stress-strain behavior of pure acrylate resin and lignin printed composites (representative curves).

**Figure 6 polymers-14-04164-f006:**
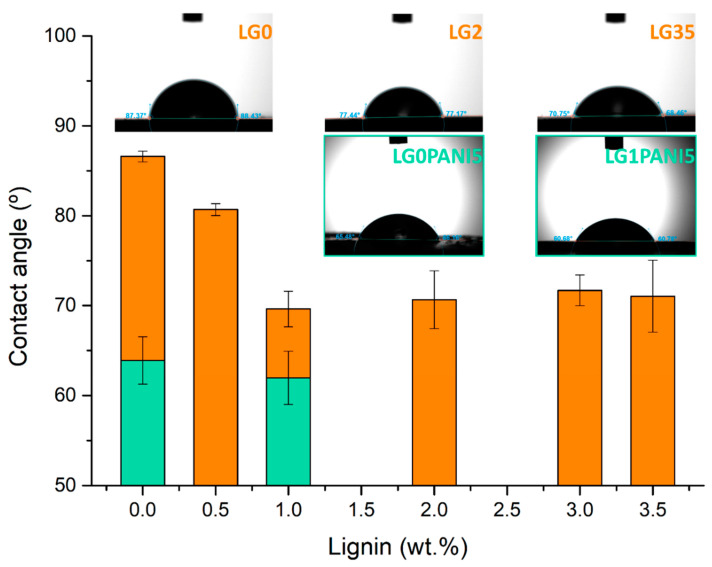
Water contact angle measurements on neat resin (L0), lignin composites of increasing lignin content (orange), and pTSA-PANIi composites (L0PANI5 and L1PANI5) (green).

**Figure 7 polymers-14-04164-f007:**
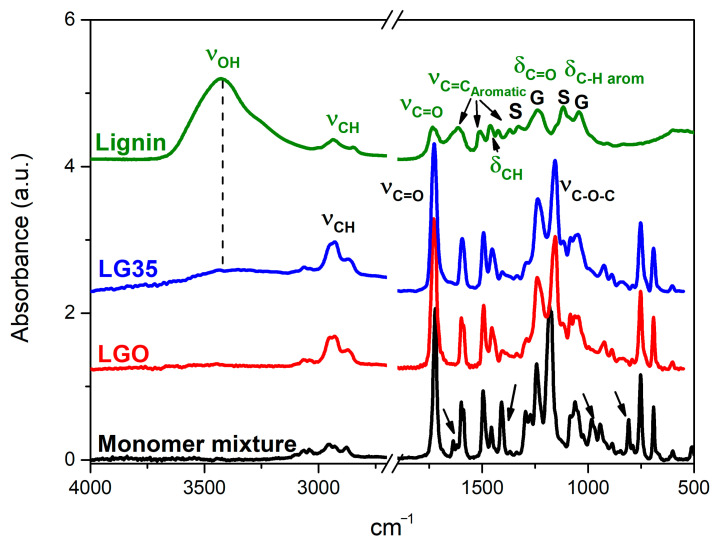
FTIR spectra of pTSA-PANI, pristine acrylic (LG0) and the composites LG0PANI5 and LG1PANI5.

**Figure 8 polymers-14-04164-f008:**
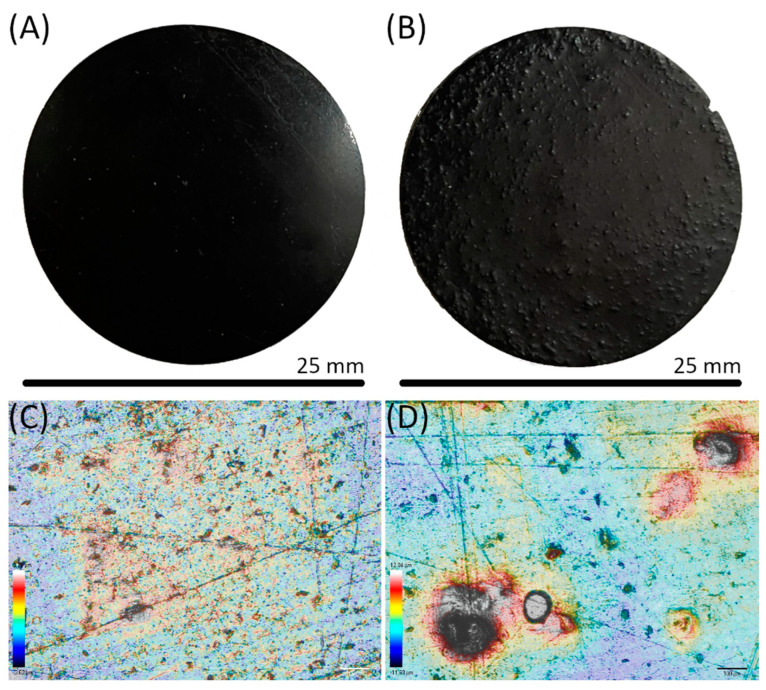
Digital pictures of printed films of (**A**) LG1PANI5 and (**B**) LG0PANI5 and representative confocal microscope images of (**C**) LG1PANI5 and (**D**) LG0PANI5.

**Table 1 polymers-14-04164-t001:** Samples formulation, 3D printer exposure times, viscosity at 1 s^−1^, maximum rate of polymerization, and ultimate degree of conversion at 9 mW cm^−2^ obtained from Real-Time ATR-FTIR spectroscopy.

Sample	Lignin (wt.%)	Exposure Time (s)	Bottom Exposure (s)	Viscosity (Pa.s) 1 s^−1^	Induction Period (s)	Max. Rate of Polymerization (mol L^−1^ s^−1^)	Ultimate DBC_∞_ (%)
LG0	0	1	15	0.011	<8	62.0 ± 0.6	98.0 ± 0.2
LG05	0.5	2	15	0.019	<8	17.9 ± 2.4	97.9 ± 1.6
LG1	1	2.5	20	0.026	24 ± 7	5.8 ± 1.8	94.4 ± 0.1
LG2	2	5	30	0.056	32 ± 13	1.9 ± 0.5	95.3 ± 0.5
LG3	3	20	50	0.060	38 ± 4	1.9 ± 0.1	94.8 ± 2.0
LG35	3.5	30	75	0.064	-	-	-
LG4	4	35	90	0.072	- *	0.9 ± 0.2	79.0 ± 11.0

* Not possible accurate determination.

**Table 2 polymers-14-04164-t002:** Characteristic temperatures measured from TGA thermograms. T_10_: temperature at which 10% of the total mass is volatilized; T_onset_: degradation temperature; DTG_max_: maximum of thermal decomposition temperature; Residue: non-volatized weight fraction at 700 °C.

Sample	T_10_ (°C)	T_onset_ (°C)	DTG_max_ (°C)	Residue (wt.%)
** *Lignin* **	208.2 ± 3.7	176.7 ± 4.0	223 and 346	35.1 ± 0.2
** *LG0* **	374.5 ± 0.6	400.0 ± 0.1	426.5 ± 3.5	2.9 ± 0.4
** *LG05* **	369.2 ± 0.2	395.6 ± 1.6	420.5 ± 2.1	3.2 ± 0.2
** *LG1* **	366.9 ± 1.6	395.5 ± 0.3	421.5 ± 0.7	3.4 ± 0.1
** *LG2* **	363.6 ± 0.8	389.4 ± 3.8	419.5 ± 0.7	3.4 ± 0.1
** *LG3* **	359.0 ± 0.2	386.5 ± 3.5	419.0 ± 1.4	3.7 ± 0.1
** *LG35* **	356.9 ± 0.9	384.8 ± 1.9	420.0 ± 1.4	3.9 ± 0.1
** *LG4* **	353.2 ± 3.0	383.8 ± 4.4	419.5 ± 0.7	4.1 ± 0.1

**Table 3 polymers-14-04164-t003:** Influence of lignin content on the mechanical properties of the printed films: Shore A hardness and tensile test parameters (E = Young’s modulus, σ = stress at break, and ε = elongation at break).

Sample	Hardness Shore A (°Sh)	E (MPa)	σ (MPa)	ε (%)
LG0	78.3 ± 1.5	4.5 ± 0.3	1.58 ± 0.26	40.6 ± 5.1
LG05	78.8 ± 0.6	4.2 ± 0.2	1.60 ± 0.10	43.9 ± 2.2
LG1	78.0 ± 1.1	3.9 ± 0.3	1.30 ± 0.13	37.9 ± 2.1
LG2	77.7 ± 1.6	4.3 ± 0.1	1.71 ± 0.11	44.13 ± 2.4
LG3	80.9 ± 1.3	5.4 ± 0.4	1.49 ± 0.20	36.1 ± 5.8
LG35	81.6 ± 0.8	5.77 ± 0.4	1.02 ± 0.30	29.2 ± 9.0
LG4	82.2 ± 1.2	6.4 ± 0.6	1.29 ± 0.30	30.4 ± 8.0

**Table 4 polymers-14-04164-t004:** R_a_ (roughness average), R_v_ (average max roughness valley depth), and R_p_ (average max roughness peak height) of the pTSA-PANI acrylic composites.

Sample	R_a_	R_v_	R_p_
PANI5LG0	3.6 ± 0.5	80.1 ± 9.9	72.5 ± 16.3
PANI5LG1	1.4 ± 0.2	32.2 ± 6.2	51.5 ± 26.5

**Table 5 polymers-14-04164-t005:** Recent research on light-curable 3D printed resins with electrical properties based on PANI.

Printing Technique	Filler	Loading (wt.%)	Base Matrix	Conductivity (S cm^−1^)	Ref
DLP	graphene sheets/PANI	1.2/5	Polyacrylate resin	4.0 × 10^−9^	[78]
DLP	PANI	5	Polyacrylate resin	1.0 × 10^−10^	[78]
DLP	PANI	6	Polyurethane	9.3 × 10^−7^	[79]
DLP	PANI	3	Acrylic resin	2.2 × 10^−5^	[31]
DLP	PANI/MWCNT	2	Acrylic resin	7.4 × 10^−4^	[32]
DLP	pTSA-PANI	5	Acrylic resin	1.7 × 10^−7^	Present work
DLP	pTSA-PANI/LIGNIN	5/1	Acrylic resin	1.6 × 10^−6^	Present work

## Data Availability

The data presented in this study are available on request from the corresponding author.

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
