# Peer review of "Lignin as a High-Value Bioaditive in 3D-DLP Printable Acrylic Resins and Polyaniline Conductive Composite"

_polymers, 2022, doi:10.3390/polym14194164_

Round 1

Reviewer 1 Report

The paper you wish to submit to Polymers is interesting and I propose it for publication after addressing the following highlights:

1. In the introduction, you have talked about the application of different applicable materials, however, I think it is better if you can consider a brief explanation through the the different additive manufacturing process. Please kindly consider the following references as well:

https://doi.org/10.3390/polym14173674

2. Also, I propose to consider a double-check on the grammatical issues.

3. The sections and sub-sections are not defined well. Please consider revising.

4. Figure 1: the quality of the figure is not good and it should be completely clear.

5. As mentioned, the sections are not well defined, and the 'material and method' should be the second section, and the 'results and discussion' the third section, and so on.

6. Figure 3, 4: Consider increasing the scale bar, it is not clear.

Best wishes

Author Response

September 29, 2022

Editorial Office

Polymers

Comments from the editors and reviewers:

Dear Editor,

Thanks a lot for considering our paper worth of being published after revision.

We also thank the referees for the constructive feedback. All suggestions, comments, and recommendations from the reviewers, have been carefully taken into account and the manuscript has been modified accordingly. Further, some other corrections have been implemented to improve the whole manuscript.

In the following, the answers to the reviewers´ comments, as well as the main changes of the text, which have been highlighted in green are shown.

We truly believe that with these clarifications and with the modifications in the manuscript, the paper is certainly improved, and meets the requirements for publication in Polymers.

Reviewer #1: The paper you wish to submit to Polymers is interesting and I propose it for publication after addressing the following highlights:

  1. the introduction, you have talked about the application of different applicable materials, however, I think it is better if you can consider a brief explanation through the the different additive manufacturing process. Please kindly consider the following references as well: https://doi.org/10.3390/polym14173674

The reference has been included in the introduction to provide the reader with a broader view of the additive manufacturing techniques as recommended.

  1. Also, I propose to consider a double-check on the grammatical issues.

The authors appreciate the comment and the writing and spelling have been revised.

  1. The sections and sub-sections are not defined well. Please consider revising.

There was an error in the numbering of the sections of the manuscript. This error has been corrected.

  1. Figure 1: the quality of the figure is not good and it should be completely clear.

Following the reviewer's suggestions, the figure has been modified and the definition and quality of the figure has been increased.

  1. As mentioned, the sections are not well defined, and the 'material and method' should be the second section, and the 'results and discussion' the third section, and so on.

This error has been corrected in the original manuscript.

  1. Figure 3, 4: Consider increasing the scale bar, it is not clear.

The scale bar has been increased in figures 3 and 4, following the reviewer's indications.

Reviewer 2 Report

Abad et al. investigated the effect of unmodified lignin content on the properties of 3D-printed light-curable acrylic composites. The results suggest that the lignin/pTSA-PANI/acrylic composite showed a clear improvement in the dispersion of the conductive filler, reducing the average surface roughness (Ra) by 61% and increasing the electrical conductivity by an order of magnitude (up to 10-6 S cm-1) compared to lignin free PANI composites. The present research opens the possibility of developing a range of novel solvent-free, eco-friendly photocurable nanocomposites for the fabrication of functional materials at low costs, valorizing a natural resource such as lignin. The novelty is clear, and this work is interesting for the readers of Polymers. However, the following comments should be addressed before the manuscript can be published.

1.      The authors should provide the tensile stress-strain curves for the samples. In addition, the authors should provide the dimensions of the samples for the tensile tests in the manuscript.

2.      Can the authors provide the particle size distribution of the lignin they used?

3.      The lignin particles are less than 100 nm in diameter. To observe the distribution of lignin in the polymer PANI matrix, the authors should include some high-resolution SEM images (e.g. 20,000 X) in the manuscript.

4.      Would the irregular distribution of lignin particles in the polymer matrix lead to heterogeneity in the properties of the final composites?

5.      Can the authors move the TGA curves of the composites to the main text?

6.      Does the improved wettability reduce the water or moisture stability of pTSA-PANIi composites?

7.      Could the authors use a profilometer or atomic force microscopy (AFM) to characterize the surface roughness of the composites?

8.      The authors should use a table or graph to compare the electrical conductivity of the composites with other previously reported similar materials.

9.      There are some grammatical/spelling errors that the author should correct, for example, on page 1, line 17; page 2, line 71; page 14, line 541…

Author Response

Reviewer #2: Abad et al. investigated the effect of unmodified lignin content on the properties of 3D-printed light-curable acrylic composites. The results suggest that the lignin/pTSA-PANI/acrylic composite showed a clear improvement in the dispersion of the conductive filler, reducing the average surface roughness (Ra) by 61% and increasing the electrical conductivity by an order of magnitude (up to 10-6 S cm-1) compared to lignin free PANI composites. The present research opens the possibility of developing a range of novel solvent-free, eco-friendly photocurable nanocomposites for the fabrication of functional materials at low costs, valorizing a natural resource such as lignin. The novelty is clear, and this work is interesting for the readers of Polymers. However, the following comments should be addressed before the manuscript can be published.

  1. The authors should provide the tensile stress-strain curves for the samples. In addition, the authors should provide the dimensions of the samples for the tensile tests in the manuscript.

The tensile stress-strain curves have been moved from the Supporting Information document to the main manuscript (Figure 5) and the dimensions of the dog-bone samples were added.

Figure 5. (A) TGA curves of Lignin and printed composites LG0, LG2 and (B) Stress-strain behavior of pure acrylate resin and lignin printed composites (representative curves).

“Tensile stress–strain mechanical properties were characterized using an Instron 5569 universal testing machine (Instron Canton, MA). The analysis was performed using a cross-head speed of 5 mm min-1 until failure, at room temperature. At least five dog-bone shaped specimens were tested following ISO 527 (dimensions 75 x 13 x 2 mm; width of narrow section 5 mm).” page 5

  1. Can the authors provide the particle size distribution of the lignin they used?

The size of the particles has been determined by SEM (Figure S2). The particle size distribution was not calculated at this stage taking into account that the unmodified lignin tends to form aggregates in the acrylic resin, as indicated in the manuscript. We consider that this kind of study will be of greater interest in the next step of the research, where the lignin will be functionalized to avoid aggregation. 

  1. The lignin particles are less than 100 nm in diameter. To observe the distribution of lignin in the polymer PANI matrix, the authors should include some high-resolution SEM images (e.g. 20,000 X) in the manuscript.

An additional figure x10000 has been included in the supplementary material (Figure S6) as suggested by the referee. The text in the manuscript has been modified accordingly.

“These images prove the little interaction between the acrylic matrix and the conductive charge in the absence of lignin, whereas the addition of lignin improves the embedding of PANI in the matrix, as shown in more detailed in Figure S6.” page 17

  1. Would the irregular distribution of lignin particles in the polymer matrix lead to heterogeneity in the properties of the final composites?

Indeed, this is one of the main problems derived from the use of unmodified lignin, which tends to form aggregates since it does not interact with the acrylic resin. This leads to poor dispersion, heterogeneous samples and poor mechanical properties. This problem will be addressed in the next step of the research in which the lignin will be modified with acrylate groups to improve dispersion and to be able to introduce larger amounts of lignin into the polymer.

  1. Can the authors move the TGA curves of the composites to the main text?

The TGA curves have been moved to the main text (Figure 5A).

  1. Does the improved wettability reduce the water or moisture stability of pTSA-PANIi composites?

Water sorption and moisture stability studies have not been carried out in this research as pTSA-PANI composites were only evaluated as a way of example to show the potential of lignin to disperse the conductive filler. However, our experience with acrylic composites (with this particular polymer matrix whose basic monomer is greatly hydrophobic) with 5 wt.% HCl-PANI confirms that these low PANI contents barely affect these bulk properties; being pTSA dopant more hydrophobic than HCl a reduction in moisture stability is not expected.

  1. Could the authors use a profilometer or atomic force microscopy (AFM) to characterize the surface roughness of the composites?

The reason why the aforementioned techniques have not been used is that, although they are very accurate measuring the surface roughness of small areas (micrometers), they are more adequate for more homogenous samples. Due to the heterogeneity of the samples, Confocal microscopy has been selected to calculate the average roughness of a greater area.

  1. The authors should use a table or graph to compare the electrical conductivity of the composites with other previously reported similar materials.

Table 4 has been added to compare the electrical conductivity of the pTSA-PANI composites developed in this work with that of similar PANI-based materials previously published.

The content added to the original manuscript is shown below.

“Regarding the incorporation of PANI as conductive filler in photocurable resins, very few studies exist, and the results obtained show medium- low conductivity values. Table 4 summarizes the main recent research on light-curable 3D printed resins with electrical properties based on PANI. Although PANI/MWCNT has been extensively studied in the field of electronics, only one previous research used this filler in a photocurable resin for SLA/DLP.”. Page 18

Table 4. Recent research on light-curable 3D printed resins with electrical properties based on PANI.

Printing Technique

Filler

Loading

(wt.%)

Base matrix

Conductivity

(S cm-1)

Ref

DLP

graphene sheets/PANI

1.2/5

polyacrylate resin

4.0 x 10-9

[78]

DLP

PANI

5

polyacrylate resin

1.0 x 10-10

[78]

DLP

PANI

6

polyurethane

9.3 x 10-7

[79]

DLP

PANI

3

Acrylic resin

2.2 x 10-5

[80]

DLP

PANI/MWCNT

2

Acrylic resin

7.4 x 10-4

[37]

DLP

pTSA-PANI

5

Acrylic resin

1.7 x 10-7

Present work

DLP

pTSA-PANI/ LIGNIN

5/1

Acrylic resin

1.6 x 10-6

Present work

  1. There are some grammatical/spelling errors that the author should correct, for example, on page 1, line 17; page 2, line 71; page 14, line 541…

The authors are grateful for the referee's recommendation and the manuscript has been checked for grammatical and spelling errors.
